# Unifying Aspects of Generalized Calculus

**DOI:** 10.3390/e22101180

**Published:** 2020-10-19

**Authors:** Marek Czachor

**Affiliations:** Wydział Fizyki Technicznej i Matematyki Stosowanej, Politechnika Gdańska, 80-233 Gdańsk, Poland; mczachor@pg.edu.pl

**Keywords:** non-Newtonian calculus, non-Diophantine arithmetic, Kolmogorov–Nagumo averages, escort probabilities, generalized entropies

## Abstract

Non-Newtonian calculus naturally unifies various ideas that have occurred over the years in the field of generalized thermostatistics, or in the borderland between classical and quantum information theory. The formalism, being very general, is as simple as the calculus we know from undergraduate courses of mathematics. Its theoretical potential is huge, and yet it remains unknown or unappreciated.

## 1. Introduction

Studies of a calculus based on generalized forms of arithmetic were initiated in the late 1960s by Grossman and Katz, resulting in their little book *Non-Newtonian Calculus* [1,2,3]. Some twenty years later, the main construction was independently discovered in a different context and pushed in a different direction by Pap [4,5,6]. After another two decades the same idea, but in its currently most general form, was rediscovered by myself [7,8,9,10,11,12,13,14,15]. In a wider perspective, non-Newtonian calculus is conceptually related to the works of Rashevsky [16] and Burgin [17,18,19,20] on non-Diophantine arithmetics of natural numbers, and to Benioff’s attempts [21,22,23,24,25] of basing physics and mathematics on a common fundamental ground. Traces of non-Newtonian and non-Diophantine thinking can be found in the works of Kaniadakis on generalized statistics [26,27,28,29,30,31,32,33,34]. A relatively complete account of the formalism can be found in the forthcoming monograph [35].

In the paper, we will discuss links between generalized arithmetics; non-Newtonian calculus; generalized entropies; and classical, quantum, and escort probabilities. As we will see, certain constructions such as Rényi entropies or exponential families of probabilities have direct relations to generalized arthmetics and calculi. Some of the constructions one finds in the literature are literally non-Newtonian. Some others only look non-Newtonian, but closer scrutiny reveals formal inconsistencies, at least from a strict non-Newtonian perspective.

Our goal is to introduce non-Newtonian calculus as a sort of unifying principle, simultaneously sketching new theoretical directions and open questions.

## 2. Non-Diophantine Arithmetic and Non-Newtonian Calculus

The most general form of non-Newtonian calculus deals with functions *A* defined by the commutative diagram ($f_{X}$ and $f_{Y}$ are arbitrary bijections)
(1)$$\begin{array}{c} \begin{array}{ccc} X & \overset{A}{⟶} & Y\\ f_{X}↓ & & ↓f_{Y}\\ R & \overset{\tilde{A}}{⟶} & R \end{array} \end{array}$$

The only assumption about the domain X and the codomain Y is that they have the same cardinality as the continuum R. The latter guarantees that bijections $f_{X}$ and $f_{Y}$ exist. The bijections are automatically continuous in the topologies they induce from the open-interval topology of R, even if they are discontinuous in metric topologies of X and Y (a typical situation in fractal applications, or in cases where X or Y are not subsets of R). In general, one does *not* assume anything else about $f_{X}$ and $f_{Y}$. In particular, their differentiability in the usual (Newtonian) sense is not assumed. No topological assumptions are made about X and Y. Of course, the structure of the diagram implies that X and Y may be regarded as Banach manifolds with global charts $f_{X}$ and $f_{Y}$, but one does not make the usual assumptions about changes of charts.

Non-Newtonian calculus begins with (generalized, non-Diophantine) arithmetics in X and Y, induced from R,
(2)$$\begin{array}{ccc} x_{1}{⊕}_{X}x_{2} & = & f_{X}^{-1}\left(f_{X}\left(x_{1}\right)+f_{X}\left(x_{2}\right)\right), \end{array}$$
(3)$$\begin{array}{ccc} x_{1}{⊖}_{X}x_{2} & = & f_{X}^{-1}\left(f_{X}\left(x_{1}\right)-f_{X}\left(x_{2}\right)\right), \end{array}$$
(4)$$\begin{array}{ccc} x_{1}{⊙}_{X}x_{2} & = & f_{X}^{-1}\left(f_{X}\left(x_{1}\right)\cdot f_{X}\left(x_{2}\right)\right), \end{array}$$
(5)$$\begin{array}{ccc} x_{1}{⊘}_{X}x_{2} & = & f_{X}^{-1}\left(f_{X}\left(x_{1}\right)/f_{X}\left(x_{2}\right)\right) \end{array}$$
(and analogously in Y).

**Example** **1.**
*According to one of the axioms of standard quantum mechanics, states of a quantum system belong to a separable Hilbert space. All separable Hilbert spaces are isomorphic, so state spaces of any two quantum systems are isomorphic. Does it mean that all quantum systems are equivalent? No, it only shows that mathematically isomorphic structures can play physically different roles. Similarly, the arithmetic given by (Equation 2)–(Equation 5) is isomorphic to the standard arithmetic of R, but it does not imply that the two arithmetics are physically equivalent.*


**Example** **2.**
*The origin of Einstein’s special theory of relativity goes back to the observation that the velocity of a source of light does not influence the velocity of light itself, contradicting our everyday experiences with velocities in trains or football. Relativistic addition of velocities is based on a fundamental unit c and the dimensionless parameter β, related to velocity by v=βc. β∈X=(−1,1) while the bijection reads $f_{X}\left(\beta \right)=\mathrm{arctanh}\beta$. The velocities are added or subtracted by means of (Equation 2) and (Equation 3),*
(6)$$\begin{array}{c} \beta_{1}{⊕}_{X}\beta_{2}=\tanh\left(\mathrm{arctanh}\beta_{1}+\mathrm{arctanh}\beta_{2}\right). \end{array}$$

*Interestingly, (Equation 4) and (Equation 5) are not directly employed in special relativity. The presence of the fundamental unit c is a signature of a general non-Diophantine arithmetic (which typically works with dimensionless numbers). Numbers ±1∈R play the roles of infinities, $\pm 1_{R}=\pm \infty_{X}$. The velocity of light is therefore literally infinite in the non-Diophantine sense. The neutral element of multiplication, $1_{X}=f_{X}^{-1}\left(1\right)=\tanh1=0.76$ (i.e., v=0.76c), does not seem to play in relativistic physics any privileged role.*


Sometimes, for example in the context of Bell’s theorem, one works with mixed arithmetics of the form [13]
(7)$$\begin{array}{c} x_{1}{⊙}_{Z}^{\mathrm{XY}}y_{2}=f_{Z}^{-1}\left(f_{X}\left(x_{1}\right)\cdot f_{Y}\left(y_{2}\right)\right),\;{⊙}_{Z}^{\mathrm{XY}}:X\times Y\to Z,\;\mathrm{etc}. \end{array}$$

Mixed arithmetics naturally occur in Taylor expansions of functions whose domains and codomains involve different arithmetics, and in the chain rule for derivatives (see Example 6).

In order to define calculus one needs limits “to zero”, and thus the notion of zero itself. In the arithmetic context a zero is a neutral element of addition, for example, $x{⊕}_{X}0_{X}=x$ for any x∈X. Obviously, such a zero is arithmetic-dependent. The same concerns a “one”, a neutral element of multiplication, fulfilling $x{⊙}_{X}1_{X}=x$ for any x∈X. Once the arithmetic in X is specified, both neutral elements are uniquely given by the general formula: $r_{X}=f_{X}^{-1}\left(r\right)$ for any r∈R. Therefore, in particular, $0_{X}=f_{X}^{-1}\left(0\right)$, $1_{X}=f_{X}^{-1}\left(1\right)$. One easily verifies that
(8)$$\begin{array}{ccc} r_{X}⊕s_{X} & = & {\left(r+s\right)}_{X}, \end{array}$$
(9)$$\begin{array}{ccc} r_{X}⊙s_{X} & = & {\left(rs\right)}_{X}, \end{array}$$
for all r,s∈R, which extends also to mixed arithmetics,
(10)$$\begin{array}{ccc} r_{X}{⊕}_{X}^{\mathrm{XY}}s_{Y} & = & {\left(r+s\right)}_{X}, \end{array}$$
(11)$$\begin{array}{ccc} r_{X}{⊕}_{Y}^{\mathrm{XY}}s_{Y} & = & {\left(r+s\right)}_{Y}, \end{array}$$
(12)$$\begin{array}{ccc} r_{X}{⊕}_{Z}^{\mathrm{XY}}s_{Y} & = & {\left(r+s\right)}_{Z},\;\mathrm{etc}. \end{array}$$

If there is no danger of ambiguity one can simplify the notation by ${⊕}_{X}^{\mathrm{XX}}={⊕}_{X}$ or ${⊕}_{X}^{\mathrm{XY}}={⊕}_{X}^{Y}$. Mixed arithmetics can be given an interpretation in terms of communication channels. Mixed multiplication is in many respects analogous to a tensor product [13].

**Example** **3.**
*Consider $X=R_{+}$, $Y=-R_{+}$, $f_{X}\left(x\right)=\ln x$, $f_{X}^{-1}\left(r\right)=e^{r}$, $f_{Y}\left(x\right)=\ln\left(-x\right)$, $f_{Y}^{-1}\left(r\right)=-e^{r}$. “Two plus two equals four” looks here as follows,*
(13)$$\begin{array}{ccc} 2_{X}{⊕}_{X}2_{X} & = & f_{X}^{-1}\left(2+2\right)=4_{X}=e^{4}, \end{array}$$
(14)$$\begin{array}{ccc} 2_{X}{⊕}_{X}^{Y}2_{Y} & = & f_{X}^{-1}\left(2+2\right)=4_{X}=e^{4}, \end{array}$$
(15)$$\begin{array}{ccc} 2_{Y}{⊕}_{Y}2_{Y} & = & f_{Y}^{-1}\left(2+2\right)=4_{Y}=-e^{4}, \end{array}$$
(16)$$\begin{array}{ccc} 2_{X}{⊕}_{Y}^{X}2_{Y} & = & f_{Y}^{-1}\left(2+2\right)=4_{Y}=-e^{4}, \end{array}$$
*where $2_{X}=f_{X}^{-1}\left(2\right)=e^{2}$, $2_{Y}=f_{Y}^{-1}\left(2\right)=-e^{2}$. From the point of view of communication channels the situation is as follows. There are two parties (“Alice” and “Bob”), each computing by means of her/his own rules. They communicate their results and agree the numbers they have found are the same, namely, “two” and “four”. However, for an external observer (an eavesdropper “Eve”), their results are opposite, say $e^{4}$ and $-e^{4}$. Mixed arithmetic plays a role of a “connection” relating different local arithmetics. This is why, in the terminology of Burgin, these types or arithmetics are non-Diophantine (from Diophantus of Alexandria who formalized the standard arithmetic). Similarly to nontrivial manifolds, non-Diophantine arithmetics do not have to admit a single global description (which we nevertheless assume in this paper).*


A limit such as $\lim_{x^{\prime }\to x}A\left(x^{\prime }\right)=A\left(x\right)$ is defined by the diagram (Equation 1) as follows,
(17)$$\begin{array}{c} \lim_{x^{\prime }\to x}A\left(x^{\prime }\right)=f_{Y}^{-1}\left(\lim_{r\to f_{X}\left(x\right)}\tilde{A}\left(r\right)\right) \end{array}$$
i.e., in terms of an ordinary limit in R. A non-Newtonian derivative is then defined by
(18)$$\begin{array}{ccc} \frac{DA(x)}{Dx} & = & \lim_{\delta \to 0}\left(A\left(x{⊕}_{X}\delta_{X}\right){⊖}_{Y}A\left(x\right)\right){⊘}_{Y}\delta_{Y}=f_{Y}^{-1}\left(\frac{d\tilde{A}\left(f_{X}\left(x\right)\right)}{df_{X}\left(x\right)}\right), \end{array}$$
if the Newtonian derivative $d\tilde{A}\left(r\right)/dr$ exists. It is additive,
(19)$$\begin{array}{ccc} \frac{D[A\left(x\right){⊕}_{Y}B\left(x\right)]}{Dx} & = & \frac{DA(x)}{Dx}{⊕}_{Y}\frac{DB(x)}{Dx}, \end{array}$$
and satisfies the Leibniz rule,
(20)$$\begin{array}{ccc} \frac{D[A\left(x\right){⊙}_{Y}B\left(x\right)]}{Dx} & = & \left(\frac{DA(x)}{Dx}{⊙}_{Y}B\left(x\right)\right){⊕}_{Y}\left(A\left(x\right){⊙}_{Y}\frac{DB(x)}{Dx}\right). \end{array}$$

A general chain rule for compositions of functions involving arbitrary arithmetics in domains and codomains can be derived [12] (see Example 6). It implies, in particular, that the bijections defining the arithmetics are themselves always non-Newtonian differentiable (with respect to the derivatives they define). The resulting derivatives are “trivial”,
(21)$$\begin{array}{c} \frac{Df_{X}\left(x\right)}{Dx}=1=\frac{Df_{Y}\left(y\right)}{Dy},\;\frac{Df_{X}^{-1}\left(r\right)}{Dr}=1_{X},\;\frac{Df_{Y}^{-1}\left(r\right)}{Dr}=1_{Y}. \end{array}$$

A non-Newtonian integral is defined by the requirement that, under typical assumptions paralleling those from the fundamental theorem of Newtonian calculus, one finds
(22)$$\begin{array}{ccc} \frac{D}{Dx}\int_{y}^{x}A\left(x^{\prime }\right)Dx^{\prime } & = & A(x), \end{array}$$
(23)$$\begin{array}{ccc} \int_{y}^{x}\frac{DA(x^{\prime })}{Dx^{\prime }}Dx^{\prime } & = & A\left(x\right){⊖}_{Y}A\left(y\right), \end{array}$$
which uniquely implies that
(24)$$\begin{array}{ccc} \int_{y}^{x}A\left(x^{\prime }\right)Dx^{\prime } & = & f_{Y}^{-1}\left(\int_{f_{X}\left(y\right)}^{f_{X}\left(x\right)}\tilde{A}\left(r\right)dr\right). \end{array}$$

Here, as before, $\tilde{A}$ is defined by (Equation 1) and dr denotes the usual Newtonian (Riemann, Lebesgue, etc.) integration. To have a feel of the potential inherent in this simple formula, let us mention that for a Koch-type fractal (Equation 24) turns out to be equivalent to the Hausdorff integral [12,36,37]. In applications, typically the only nontrivial element is to find the explicit form of $f_{X}$. It should be stressed that (Equation 24) reduces any integral to the one over a subset of R. The fact that such a counterintuitive possibility exists was noticed already by Wiener in his 1933 lectures on Fourier analysis [38].

## 3. Non-Newtonian Exponential Function and Logarithm

Once we know how to differentiate and integrate, we can turn to differential equations. The so-called exponential family plays a crucial role in thermodynamics, both standard and generalized [39,40,41,42,43]. Many different deformations of the usual $e^{x}$ can be found in the literature. However, from the non-Newtonian perspective, the exponential function Exp:X→Y is defined by
(25)$$\begin{array}{c} \frac{D\mathrm{Exp}(x)}{Dx}=\mathrm{Exp}\left(x\right),\;\mathrm{Exp}\left(0_{X}\right)=1_{Y}. \end{array}$$

Integrating (Equation 25) (in a non-Newtonian way) one finds the unique solution
(26)$$\begin{array}{c} \mathrm{Exp}\left(x\right)=f_{Y}^{-1}\left(e^{f_{X}\left(x\right)}\right),\;\mathrm{Exp}\left(x_{1}{⊕}_{X}x_{2}\right)=\mathrm{Exp}\left(x_{1}\right){⊙}_{Y}\mathrm{Exp}\left(x_{2}\right). \end{array}$$

In thermodynamic applications, one often encounters exponents of negative arguments, $e^{-x}$. In a non-Newtonian context the correct form of a minus is ${⊖}_{X}x=0_{X}{⊖}_{X}x=f_{X}^{-1}\left(-f_{X}\left(x\right)\right)$. The example discussed in the next section will involve X=R and $f_{X}^{-1}\left(-r\right)=-f_{X}^{-1}$(r). In consequence, it will be correct to write ${⊖}_{X}x=-x$, but in general such a simple rule may be meaningless (because “−”, as opposed to ${⊖}_{X}$, may be undefined in X).

**Example** **4.**
*Let $X=(R_{+},⊕,⊙\;)$, with the arithmetic defined by $f_{X}:R_{+}\to R$, $f_{X}\left(x\right)=\ln x$, $f_{X}^{-1}\left(r\right)=e^{r}$. Then*
(27)$$\begin{array}{c} {⊖}_{X}x=f_{X}^{-1}\left(-f_{X}\left(x\right)\right)=e^{-\ln x}=1/x\in R_{+}. \end{array}$$

*The same number can be both positive and negative, depending on the arithmetic.*


A (natural) logarithm is the inverse of Exp, namely, Ln:Y→X,
(28)$$\begin{array}{c} \mathrm{Ln}\left(y\right)=f_{X}^{-1}\left(\ln f_{Y}\left(x\right)\right),\;\mathrm{Ln}\left(y_{1}{⊙}_{Y}y_{2}\right)=\mathrm{Ln}\left(y_{1}\right){⊕}_{X}\mathrm{Ln}\left(y_{2}\right). \end{array}$$

Expressions such as Expx+Lny are in general meaningless even if $X\subset R_{+}$ and $Y\subset R_{+}$. However, formulas such as
(29)$$\begin{array}{c} \left(\mathrm{Exp}x\right){⊕}_{Z}^{YX}\left(\mathrm{Ln}y\right)=f_{Z}^{-1}\left(e^{f_{X}\left(x\right)}+\ln f_{Y}\left(y\right)\right) \end{array}$$
make perfect sense. For example, if $p_{k}\in X$, then an entropy can be defined as
(30)$$\begin{array}{ccc} S & = & \underset{k}{⨁}{}_{Z}p_{k}{⊙}_{Z}^{XY}\mathrm{Ln}\left(1_{X}{⊘}_{X}p_{k}\right) \end{array}$$
(31)$$\begin{array}{ccc} & = & f_{Z}^{-1}\left[\sum_{k}f_{Z}\left(p_{k}{⊙}_{Z}^{XY}\mathrm{Ln}\left(1_{X}{⊘}_{X}p_{k}\right)\right)\right] \end{array}$$
(32)$$\begin{array}{ccc} & = & f_{Z}^{-1}\left[\sum_{k}f_{X}\left(p_{k}\right)\ln\left(1/f_{X}\left(p_{k}\right)\right)\right]. \end{array}$$

Many intriguing questions occur if one asks about normalization of probabilities. We will come to it later.

Non-Newtonian constructions of Exp and Ln are systematic, general, and flexible. There seems to exist a relation between the arithmetic formalism and the method of monotone embedding discussed in information geometry [44], but the problem requires further studies.

**Example** **5.**
*In order to appreciate the difference between Newtonian and non-Newtonian differentiation let us differentiate the function A(x)=x, A:X→Y, but in two cases. The first one is trivial, X=Y=(R,+,·), with the arithmetic defined by the identity $f_{X}=f_{Y}={\mathrm{id}}_{R}$. Then, the non-Newtonian and Newtonian derivatives coincide, so*
(33)$$\begin{array}{c} \frac{DA(x)}{Dx}=\frac{dA(x)}{dx}=1. \end{array}$$

*The second case involves, as before, the codomain Y=(R,+,·), with the arithmetic defined by the identity $f_{Y}={\mathrm{id}}_{R}$. However, as the domain we choose $X=(R_{+},⊕,⊙\;)$, with the arithmetic defined by $f_{X}:R_{+}\to R$, $f_{X}\left(x\right)=\ln x$, $f_{X}^{-1}\left(r\right)=e^{r}$. Now,*
(34)$$\begin{array}{ccc} \frac{DA(x)}{Dx} & = & \lim_{\delta \to 0}\left(A\left(x{⊕}_{X}\delta_{X}\right){⊖}_{Y}A\left(x\right)\right){⊘}_{Y}\delta_{Y}=\lim_{\delta \to 0}\frac{\left(x{⊕}_{X}f_{X}^{-1}\left(\delta \right)\right)-x}{\delta }\\ & = & \lim_{\delta \to 0}\frac{e^{\ln x+\delta }-x}{\delta }=x=A\left(x\right). \end{array}$$

*As, $0_{X}=f_{X}^{-1}\left(0\right)=e^{0}=1$, we find $A\left(0_{X}\right)=0_{X}=1=1_{Y}$, and conclude that A(x)=x, $A:R_{+}\to R$ belongs to the exponential family. Indeed,*
(35)$$\begin{array}{c} A\left(x_{1}{⊕}_{X}x_{2}\right)=x_{1}{⊕}_{X}x_{2}=e^{\ln x_{1}+\ln x_{2}}=x_{1}\cdot x_{2}=A\left(x_{1}\right){⊙}_{Y}A\left(x_{2}\right). \end{array}$$

*To understand the result, write $A\left(x\right)=f_{Y}^{-1}\left(\tilde{A}(f_{X}\left(x\right)\right)=\tilde{A}\left(\ln x\right)=x$, so that $\tilde{A}\left(r\right)=e^{r}$. Then, by the second form of derivative in (Equation 18),*
(36)$$\begin{array}{ccc} \frac{DA(x)}{Dx} & = & f_{Y}^{-1}\left(\frac{d\tilde{A}\left(f_{X}\left(x\right)\right)}{df_{X}\left(x\right)}\right)=\frac{d\;e^{f_{X}\left(x\right)}}{df_{X}\left(x\right)}=e^{f_{X}\left(x\right)}=e^{\ln x}=x. \end{array}$$

*The map A does not affect the value of x, but changes its arithmetic properties. It behaves as if it assigned a different meaning to the same word. The example becomes even more intriguing if one realizes that logarithm is known to approximately relate stimulus with sensation in real-life sensory systems (hence the logarithmic scale of decibels and star magnitudes) [35].*


**Example** **6.**
*Many calculations in thermodynamics reduce to formulas of the form*
(37)$$\begin{array}{c} dU\left(S,V\right)={\left(\frac{\partial U}{\partial S}\right)}_{V}dS+{\left(\frac{\partial U}{\partial V}\right)}_{S}dV, \end{array}$$
*being equivalent to the derivative dU(S(t),V(t))/dt of a composite function of several variables. The latter has a unique formulation in non-Newtonian calculus: One only needs to specify the arithmetics. For example, let U be a map U:S×V→U, and let S:T→S, V:T→V. Then,*
(38)$$\begin{array}{ccc} \frac{DU(S(t),V(t))}{Dt} & = & \lim_{\delta \to 0}\left(U\left((S\left(t{⊕}_{T}\delta_{T}\right),V\left(t{⊕}_{T}\delta_{T}\right)\right){⊖}_{U}U\left(S(t),V(t)\right)\right){⊘}_{U}\delta_{U}. \end{array}$$

*As*
(39)$$\begin{array}{c} \lim_{x^{\prime }\to x}A\left(x\right){⊕}_{Y}B\left(x\right)=\left(\lim_{x^{\prime }\to x}A\left(x\right)\right){⊕}_{Y}\left(\lim_{x^{\prime }\to x}B\left(x\right)\right) \end{array}$$
*(see Appendix A), we rewrite (Equation 38) as*
(40)$$\begin{array}{ccc} \frac{DU(S(t),V(t))}{Dt} & = & \lim_{\delta \to 0}\left(U\left((S\left(t{⊕}_{T}\delta_{T}\right),V\left(t{⊕}_{T}\delta_{T}\right)\right){⊖}_{U}U\left((S\left(t{⊕}_{T}\delta_{T}\right),V\left(t\right)\right)\right){⊘}_{U}\delta_{U}\\ & & {⊕}_{U}\lim_{\delta \to 0}\left(U\left((S\left(t{⊕}_{T}\delta_{T}\right),V\left(t\right)\right){⊖}_{U}U\left(S(t),V(t)\right)\right){⊘}_{U}\delta_{U}. \end{array}$$

*Under the usual assumptions about continuity of $\tilde{U}:R\times R\to R$ in*
(41)$$\begin{array}{c} \begin{array}{ccc} S\times V & \overset{U}{⟶} & U\\ f_{S}↓f_{V}↓ & & ↓f_{U}\\ R\times R & \overset{\tilde{U}}{⟶} & R \end{array}, \end{array}$$
*we reduce (Equation 40) to*
(42)$$\begin{array}{ccc} \frac{DU(S(t),V(t))}{Dt} & = & \lim_{\delta \to 0}\left(U\left((S\left(t\right),V\left(t{⊕}_{T}\delta_{T}\right)\right){⊖}_{U}U\left((S(t),V(t)\right)\right){⊘}_{U}\delta_{U}\\ & & {⊕}_{U}\lim_{\delta \to 0}\left(U\left((S\left(t{⊕}_{T}\delta_{T}\right),V\left(t\right)\right){⊖}_{U}U\left(S(t),V(t)\right)\right){⊘}_{U}\delta_{U}, \end{array}$$
*and then to two instances of the non-Newtonian chain rule,*
(43)$$\begin{array}{c} \frac{D(B\circ A)(x)}{Dx}=f_{Z}^{-1}\left[f_{Z}\left(\frac{DB\left(A(x)\right)}{DA(x)}\right)f_{Y}\left(\frac{DA(x)}{Dx}\right)\right]=\frac{DB\left(A(x)\right)}{DA(x)}{⊙}_{Z}^{\mathrm{ZY}}\frac{DA(x)}{Dx}, \end{array}$$
*valid for the composition*
(44)$$\begin{array}{c} \begin{array}{ccccc} X & \overset{A}{⟶} & Y & \overset{B}{⟶} & Z\\ f_{X}↓ & & f_{Y}↓ & & f_{Z}↓\\ R & \overset{\tilde{A}}{⟶} & R & \overset{\tilde{B}}{⟶} & R \end{array} \end{array}$$
*of maps. Finally,*
(45)$$\begin{array}{c} \frac{DU(S(t),V(t))}{Dt}=\frac{DU(S(t),V(t))}{DS(t)}{⊙}_{U}^{\mathrm{US}}\frac{DS(t)}{Dt}{⊕}_{U}\frac{DU(S(t),V(t))}{DV(t)}{⊙}_{U}^{\mathrm{UV}}\frac{DV(t)}{Dt}. \end{array}$$

*Effectively,*
(46)$$\begin{array}{c} DU\left(S,V\right)={\left(\frac{DU}{DS}\right)}_{V}{⊙}_{U}^{\mathrm{US}}DS{⊕}_{U}{\left(\frac{DU}{DV}\right)}_{S}{⊙}_{U}^{\mathrm{UV}}DV, \end{array}$$
*is the non-Newtonian formula for a differential.*


The next section shows that the above mentioned subtleties with arithmetics of domains and codomains have straightforward implications for generalized thermostatistics.

## 4. Kaniadakis κ-Calculus Versus Non-Newtonian Calculus

Kaniadakis, in a series of papers [26,27,28,29,30,31,32,33,34], developed a generalized form of arithmetic and calculus, with numerous applications to statistical physics, and beyond. In the present section, we will clarify links between his formalism and non-Newtonian calculus. As we will see, some of the results have a straightforward non-Newtonian interpretation, but not all.

Assume X=R, with the bijection $f_{X}\equiv f_{\kappa }:R\to R$ given explicitly by
(47)$$\begin{array}{ccc} f_{\kappa }\left(x\right) & = & \frac{1}{\kappa }\;\mathrm{arcsinh}\kappa x, \end{array}$$
(48)$$\begin{array}{ccc} f_{\kappa }^{-1}\left(x\right) & = & \frac{1}{\kappa }\sinh\kappa x. \end{array}$$

Kaniadakis’ κ-calculus begins with the arithmetic,
(49)$$\begin{array}{ccc} x\overset{\kappa }{⊕}y & = & f_{\kappa }^{-1}\left(f_{\kappa }\left(x\right)+f_{\kappa }\left(y\right)\right), \end{array}$$
(50)$$\begin{array}{ccc} x\overset{\kappa }{⊖}y & = & f_{\kappa }^{-1}\left(f_{\kappa }\left(x\right)-f_{\kappa }\left(y\right)\right), \end{array}$$
(51)$$\begin{array}{ccc} x\overset{\kappa }{⊙}y & = & f_{\kappa }^{-1}\left(f_{\kappa }\left(x\right)\cdot f_{\kappa }\left(y\right)\right), \end{array}$$
(52)$$\begin{array}{ccc} x\overset{\kappa }{⊘}y & = & f_{\kappa }^{-1}\left(f_{\kappa }\left(x\right)/f_{\kappa }\left(y\right)\right). \end{array}$$

As $f_{0}\left(x\right)=x$, the case κ=0 corresponds to the usual field $R_{0}=\left(R,+,\cdot \right)$, which we will shortly denote by R. The neutral element of addition, $0_{\kappa }=f_{\kappa }^{-1}\left(0\right)=0$, is the same for all κs. The neutral element of κ-multiplication is nontrivial, $1_{\kappa }=f_{\kappa }^{-1}\left(1\right)\neq 1$. The fields $R_{\kappa }=\left(R,\overset{\kappa }{⊕},\overset{\kappa }{⊙}\right)$ are isomorphic to one another due to their isomorphism with $R_{0}$,
(53)$$\begin{array}{ccc} f_{\kappa }\left(x\overset{\kappa }{⊕}y\right) & = & f_{\kappa }\left(x\right)+f_{\kappa }\left(y\right), \end{array}$$
(54)$$\begin{array}{ccc} f_{\kappa }\left(x\overset{\kappa }{⊙}y\right) & = & f_{\kappa }\left(x\right)\cdot f_{\kappa }\left(y\right). \end{array}$$

Kaniadakis defines his κ-derivative of a real function A(x) as
(55)$$\begin{array}{ccc} \frac{dA(x)}{d_{\kappa }x} & = & \lim_{\delta \to 0}\frac{A(x+\delta )-A(x)}{\left(x+\delta \right)\overset{\kappa }{⊖}x}=\frac{dA(x)}{dx}/\frac{df_{\kappa }\left(x\right)}{dx}=\frac{dA(x)}{dx}\sqrt{1+\kappa^{2}x^{2}}. \end{array}$$

We will now specify in which sense the κ-derivative is non-Newtonian. First consider a function *A*,
(56)$$\begin{array}{c} \begin{array}{ccc} R_{\kappa_{1}} & \overset{A}{⟶} & R_{\kappa_{2}}\\ f_{\kappa_{1}}↓ & & ↓f_{\kappa_{2}}\\ R & \overset{\tilde{A}}{⟶} & R \end{array} \end{array}$$

Its non-Newtonian derivative
(57)$$\begin{array}{c} \frac{DA(x)}{Dx}=\lim_{\delta \to 0}\left(A\left(x\overset{\kappa_{1}}{⊕}\delta_{\kappa_{1}}\right)\overset{\kappa_{2}}{⊖}A\left(x\right)\right)\overset{\kappa_{2}}{⊘}\delta_{\kappa_{2}}, \end{array}$$
if compared with (Equation 55), suggests $\kappa_{2}=0$. Setting $\kappa_{1}=\kappa$, $\kappa_{2}=0$, we find
(58)$$\begin{array}{c} \frac{DA(x)}{Dx}=\lim_{\delta \to 0}\frac{A\left(x\overset{\kappa }{⊕}\delta_{\kappa }\right)-A\left(x\right)}{\delta }=\lim_{\delta \to 0}\frac{A[x\overset{\kappa }{⊕}f_{\kappa }^{-1}\left(\delta \right)]-A\left(x\right)}{\delta }=\lim_{\delta \to 0}\frac{A\left(x\overset{\kappa }{⊕}\delta \right)-A\left(x\right)}{\delta }, \end{array}$$
as $f_{\kappa }^{-1}\left(\delta \right)\approx \delta$ for δ≈0. Denoting $x\overset{\kappa }{⊕}\delta =x+\delta^{\prime }$ we find $\delta =\left(x+\delta^{\prime }\right)\overset{\kappa }{⊖}x$, and
(59)$$\begin{array}{c} \frac{DA(x)}{Dx}=\lim_{\delta^{\prime }\to 0}\frac{A\left(x+\delta^{\prime }\right)-A\left(x\right)}{\left(x+\delta^{\prime }\right)\overset{\kappa }{⊖}x}, \end{array}$$
in agreement with the Kaniadakis formula. However, as a by-product of the calculation we have proved that κ-calculus is applicable only to functions mapping $R_{\kappa }$ into R. Kaniadakis exponential function satisfies
(60)$$\begin{array}{c} \frac{D\mathrm{Exp}(x)}{Dx}=\mathrm{Exp}\left(x\right),\;\mathrm{Exp}\left(0\right)=1, \end{array}$$
with $0=0_{\kappa }$, $1=1_{0}$. Accordingly,
(61)$$\begin{array}{c} \mathrm{Exp}\left(x\right)=f_{Y}^{-1}\left(e^{f_{X}\left(x\right)}\right)=e^{f_{\kappa }\left(x\right)}=e^{\frac{1}{\kappa }\;\mathrm{arcsinh}\kappa x}, \end{array}$$
which is indeed the Kaniadakis result. Recalling that $f_{Y}\left(x\right)=x$, we find the explicit form of the logarithm, $\mathrm{Ln}:R\to R_{\kappa }$,
(62)$$\begin{array}{c} \mathrm{Ln}\left(y\right)=f_{X}^{-1}\left(\ln f_{Y}\left(y\right)\right)=\frac{1}{\kappa }\sinh\left(\kappa \ln y\right), \end{array}$$
which again agrees with the Kaniadakis definition.

Yet, the readers must be hereby warned that it is *not* allowed to apply the Kaniadakis definition of derivative to Lnx. The correct non-Newtonian form is
(63)$$\begin{array}{c} \frac{D\mathrm{Ln}(y)}{Dy}=\lim_{\delta \to 0}\left(\mathrm{Ln}\left(y+\delta \right)\overset{\kappa }{⊖}\mathrm{Ln}\left(y\right)\right)\overset{\kappa }{⊘}\delta_{\kappa }=f_{X}^{-1}\left(1/f_{Y}\left(y\right)\right)=\frac{1}{\kappa }\sinh\left(\kappa /y\right), \end{array}$$
because Ln maps R into $R_{\kappa }$. Kaniadakis is aware of the subtlety and thus introduces also another derivative, meant for differentiation of inverse functions,
(64)$$\begin{array}{c} \frac{d_{\kappa }A\left(y\right)}{dy}=\lim_{u\to y}\frac{A\left(y\right)\overset{\kappa }{⊖}A\left(u\right)}{y-u}=\lim_{\delta \to 0}\frac{A\left(y+\delta \right)\overset{\kappa }{⊖}A\left(y\right)}{\delta }, \end{array}$$
a definition which, from the non-Newtonian standpoint, must be nevertheless regarded as incorrect (‘/’ should be replaced by $\overset{\kappa }{⊘}$ typical of the codomain $R_{\kappa }$). As a result,
(65)$$\begin{array}{c} \frac{d_{\kappa }\mathrm{Ln}\left(y\right)}{dy}=\frac{1}{y}\neq \frac{D\mathrm{Ln}(y)}{Dy}=\frac{1}{\kappa }\sinh\frac{\kappa }{y}. \end{array}$$

This is probably why (Equation 64), as opposed to (Equation 55), has not found too many applications.

Let us finally check what would have happened if instead of (Equation 61) one considered the exponential function mapping $R_{\kappa }$ into itself, $f_{Y}=f_{X}=f_{\kappa }$,
(66)$$\begin{array}{c} \mathrm{Exp}\left(x\right)=f_{Y}^{-1}\left(e^{f_{X}\left(x\right)}\right)=f_{\kappa }^{-1}\left(e^{f_{\kappa }\left(x\right)}\right)=\frac{1}{\kappa }\sinh\left(\kappa \;e^{\frac{1}{\kappa }\;\mathrm{arcsinh}\kappa x}\right). \end{array}$$

As in thermodynamic applications one typically encounters Exp of a negative argument, one expects that physical differences between $\mathrm{Exp}:R_{\kappa }\to R_{\kappa }$ and $\mathrm{Exp}:R_{\kappa }\to R$ should not be essential. Moreover, indeed, Figure 1 shows that both exponents lead to identical asymptotic tails.

## 5. A Cosmological Aspect of the Kaniadakis Arithmetic

Kaniadakis explored possible relativistic implications of his formalism. In particular, he noted that fluxes of cosmic rays depend on energy in a way that seems to indicate κ>0. It is therefore intriguing that essentially the same arithmetic was recently shown [14] to have links with the problem of accelerated expansion of the Universe, one of the greatest puzzles of contemporary physics.

Cosmological expansion is well described by the Friedman equation,
(67)$$\begin{array}{c} \frac{da(t)}{dt}=\sqrt{\Omega_{\Lambda }a{\left(t\right)}^{2}+\frac{\Omega_{M}}{a(t)}},\;a\left(t\right)>0, \end{array}$$
for a dimensionless scale factor a(t) evolving in a dimensionless time *t* (in units of the Hubble time $t_{H}\approx 13.58\times 10^{9}$ yr). The observable parameters are $\Omega_{M}=0.3$, $\Omega_{\Lambda }=0.7$ [45,46]. $\Omega_{\Lambda }\neq 0$ is typically interpreted as an indication of dark energy. Equation (Equation 67) is solved by
(68)$$\begin{array}{c} a\left(t\right)={\left(\sqrt{\frac{\Omega_{M}}{\Omega_{\Lambda }}}\sinh\frac{3\sqrt{\Omega_{\Lambda }}t}{2}\right)}^{2/3},\;t>0. \end{array}$$

Now assume that
(69)$$\begin{array}{c} \begin{array}{ccc} X & \overset{a}{⟶} & R\\ f_{X}↓ & & ↓f_{R}={\mathrm{id}}_{R}\\ R & \overset{\tilde{a}}{⟶} & R \end{array}, \end{array}$$
whereas the Friedman equation involves no $\Omega_{\Lambda }$,
(70)$$\begin{array}{c} \frac{Da(t)}{Dt}=\sqrt{\frac{\Omega }{a(t)}},\;a\left(t\right)>0, \end{array}$$
for some Ω. Its solution by non-Newtonian techniques reads
(71)$$\begin{array}{ccc} a(t) & = & {\left(\frac{3}{2}\sqrt{\Omega }f_{X}\left(t\right)\right)}^{2/3}, \end{array}$$
so, comparing (Equation 71) with (Equation 68), we find
(72)$$\begin{array}{c} f_{X}\left(t\right)=\frac{2}{3\sqrt{0.7}}\sqrt{\frac{\Omega_{M}}{\Omega }}\sinh\frac{3\sqrt{0.7}}{2}t=\sqrt{\frac{\Omega_{M}}{\Omega }}f_{\kappa }^{-1}\left(t\right),\;\mathrm{for}\kappa =1.255. \end{array}$$

Accelerated expansion of the Universe looks like a combined effect of non-Euclidean geometry and non-Diophantine arithmetic. The resulting dynamics is non-Newtonian in both meanings of this term.

The presence of the inverse bijection $f_{\kappa }^{-1}$ and κ>1 raises a number of interesting questions. It is related to the fundamental duality between Diophantine and non-Diophantine arithmetics. Namely, any equation of the form, say
(73)$$\begin{array}{ccc} x_{1}⊕x_{2} & = & f^{-1}\left(f\left(x_{1}\right)+f\left(x_{2}\right)\right), \end{array}$$
can be inverted by f(x)=y into
(74)$$\begin{array}{ccc} y_{1}+y_{2} & = & f\left(f^{-1}\left(y_{1}\right)⊕f^{-1}\left(y_{2}\right)\right), \end{array}$$
suggesting that it is ⊕ and not + which is the Diophantine arithmetic operation. Having two isomorphic arithmetics we, in general, do not have any criterion telling us which of the two is “normal”, and which is “generalized”.

## 6. Kolmogorov–Nagumo Averages and Non-Diophantine/Non-Newtonian Probability

Another non-Diophantine/non-Newtonian aspect that can be identified in the context of information theory and thermodynamics is implicitly present in the works of Kolmogorow, Nagumo, and Rényi. Let us recall that a Kolmogorov–Nagumo average is defined as [47,48,49,50,51,52,53,54]
(75)$$\begin{array}{ccc} {\langle a\rangle }_{f} & = & f^{-1}\left(\sum_{k}p_{k}f\left(a_{k}\right)\right). \end{array}$$

Rewriting (Equation 75) as
(76)$$\begin{array}{c} {\langle a\rangle }_{f}=f^{-1}\left(\sum_{k}f\left(p_{k}^{\prime }\right)f\left(a_{k}\right)\right)=\underset{k}{⨁}p_{k}^{\prime }⊙a_{k}, \end{array}$$
where $p_{k}^{\prime }=f^{-1}\left(p_{k}\right)$, one interprets the average as the one typical of a non-Diophantine-arithmetic-valued probability. Apparently, neither Kolmogorov nor Nagumo nor Rényi had interpreted their results from this arithmetic point of view [7].

The lack of arithmetic perspective is especially visible in the works of Rényi [49] who, while deriving his α-entropies, began with a general Kolmogorov–Nagumo average. Trying to derive a meaningful class of *f*s he demanded that
(77)$$\begin{array}{c} {\langle a+c\rangle }_{f}={\langle a\rangle }_{f}+c \end{array}$$
be valid for any constant random variable *c*, and this led him to the exponential family $f_{\alpha }\left(x\right)=2^{(1-\alpha )x}$ (up to a general affine transformation f↦Af+B, which does not affect Kolmogorov–Nagumo averages). In physical applications, it is more convenient to work with natural logarithms, so let us replace $f_{\alpha }$ by $f_{q}\left(x\right)=e^{(1-q)x}$, $f_{q}^{-1}\left(x\right)=\frac{1}{1-q}\ln x$, q∈R. With this particular choice of *f* one finds
(78)$$\begin{array}{ccc} {\langle a\rangle }_{f_{q}} & = & \frac{1}{1-q}\ln\left(\sum_{k}p_{k}e^{\left(1-q\right)a_{k}}\right). \end{array}$$

As is well known, the standard linear average is the limiting case $\lim_{q\to 1}{\langle a\rangle }_{f_{q}}=\sum_{k}p_{k}a_{k},$ that includes the entropy of Shannon, $S=\sum_{k}p_{k}\ln\left(1/p_{k}\right)=S_{1}$, as the limit q→1 of the Rényi entropy
(79)$$\begin{array}{c} S_{q}=\frac{1}{1-q}\ln\left(\sum_{k}p_{k}e^{\left(1-q\right)\ln\left(1/p_{k}\right)}\right)=\frac{1}{1-q}\ln\sum_{k}p_{k}^{q}. \end{array}$$

Still, notice that ${\langle a⊕b\rangle }_{f}={\langle a\rangle }_{f}⊕{\langle b\rangle }_{f}$ for any *f*, so had Rényi been thinking in arithmetic categories, he would not have arrived at his $f_{\alpha }$. Yet, $f_{\alpha }$ is an interesting special case. For example,
(80)$$\begin{array}{c} p_{k}^{\prime }=f_{q}^{-1}\left(p_{k}\right)=\frac{1}{q-1}\ln\left(1/p_{k}\right). \end{array}$$

The random variable $a_{k}=\log_{b}\left(1/p_{k}\right)$ is, according to Shannon [49,55], the amount of information obtained by observing an event whose probability is $p_{k}$. The choice of *b* defines units of information. Therefore, Rényi’s non-Diophantine probability $p_{k}^{\prime }$ is the amount of information encoded in $p_{k}$.

## 7. Escort Probabilities and Quantum Mechanical Hidden Variables

Non-Diophantine arithmetics have several properties that make them analogous to sets of values of incompatible random variables in quantum mechanics. Generalized arithmetics and non-Newtonian calculi have nontrivial consequences for the problem of hidden variables and completeness of quantum mechanics.

**Example** **7.**
*Pauli matrices $\sigma_{1}$ and $\sigma_{2}$ represent random variables whose values are $s_{1}=\pm 1$ and $s_{2}=\pm 1$, respectively. However, it is not allowed to assume that $\sigma_{1}+\sigma_{2}$ represents a random variable whose possible values are $s_{1}+s_{2}=0,\pm 2$, even though an average of $\sigma_{1}+\sigma_{2}$ ia a sum of independent averages of $\sigma_{1}$ and $\sigma_{2}$. In non-Diophantine arithmetic one encounters a similar problem. In general it makes no sense to perform additions of the form $x_{X}+y_{Y}$ even if $x_{X}\in R$ and $y_{Y}\in R$. One should not be surprised if non-Diophantine probabilities turn out to be analogous to quantum probabilities, at least in some respects.*


Normalization of probability implies
(81)$$\begin{array}{c} 1_{X}=f^{-1}\left(1\right)=f^{-1}\left(\sum_{k}p_{k}\right)=f^{-1}\left(\sum_{k}f\left(p_{k}^{\prime }\right)\right)=\underset{k}{⨁}p_{k}^{\prime }. \end{array}$$

In principle, $1_{X}\neq 1$. An interesting and highly nontrivial case occurs if both $p_{k}$ and $p_{k}^{\prime }=f^{-1}\left(p_{k}\right)$ are probabilities in the ordinary sense, i.e., in addition to (Equation 81) one finds $1_{X}=1$, $0\leq p_{k}^{\prime }\leq 1$, and $\sum_{k}p_{k}^{\prime }=1$. What can be then said about *f*? We can formalize the question as follows.

**Problem** **1.**
*Find a characterization of those functions g:[0,1]→[0,1] that satisfy*
(82)$$\begin{array}{c} \sum_{k}g\left(p_{k}\right)=1,\;for\;any\;choice\;of\;probabilities\;p_{k}. \end{array}$$


In analogy to the generalized thermostatistics literature we can term $p_{k}^{\prime }=g\left(p_{k}\right)$ the escort probabilities [56,57,58]. Notice that we are *not* in interested in the trivial solution, often employed in the context of Tsallis and Rényi entropies, where $p_{k}$ is replaced by $p_{k}^{q}$ and then *renormalized*,
(83)$$\begin{array}{c} P_{k}=\frac{p_{k}^{q}}{\sum_{j}p_{j}^{q}}=g_{k}\left(p_{1},\ldots ,p_{n},\ldots \right) \end{array}$$
as $g_{k}\left(p_{1},\ldots ,p_{n},\ldots \right)\neq g\left(p_{k}\right)$ for a single function *g* of one variable. As we will shortly see, the solution of (Equation 82) turns out to have straightforward implications for the quantum mechanical problem of hidden variables, and relations between classical and quantum probabilities.

The most nontrivial result is found for binary probabilities, $p_{1}+p_{2}=1$.

**Lemma** **1.**
*$g\left(p_{1}\right)+g\left(p_{2}\right)=1$ for all $p_{1}+p_{2}=1$ if and only if*
(84)$$\begin{array}{c} g\left(p\right)=\frac{1}{2}+h\left(p-\frac{1}{2}\right) \end{array}$$
*where h(−x)=−h(x).*


**Proof.** See Appendix B. □

The lemma has profound consequences for foundations of quantum mechanics, as it allows to circumvent Bell’s theorem by non-Newtonian hidden variables. For more details the readers are referred to [13,15], but here just a few examples.

**Example** **8.**
*The trivial case g(p)=p implies h(x)=x, where 0≤p≤1 and −1/2≤x≤1/2.*


**Example** **9.**
*Consider $g\left(p\right)=\sin^{2}\frac{\pi }{2}p$. Then,*
(85)$$\begin{array}{c} h\left(x\right)=g\left(x+\frac{1}{2}\right)-\frac{1}{2}=\frac{1}{2}\sin\pi x. \end{array}$$

*Let us cross-check,*
(86)$$\begin{array}{c} g\left(p\right)+g\left(1-p\right)=\sin^{2}\frac{\pi }{2}p+\sin^{2}\frac{\pi }{2}\left(1-p\right)=\sin^{2}\frac{\pi }{2}p+\cos^{2}\frac{\pi }{2}p=1. \end{array}$$

*Now let p=(π−θ)/π be the probability of finding a point belonging to the overlap of two half-circles rotated by θ. Then,*
(87)$$\begin{array}{c} g\left(p\right)=\sin^{2}\frac{\pi }{2}\frac{\pi -\theta }{\pi }=\cos^{2}\frac{\theta }{2} \end{array}$$
*is the quantum-mechanical law describing the conditional probability for two successive measurements of spin-1/2 in two Stern–Gerlach devices placed one after another, with relative angle θ. Escort probability has become a quantum probability.*


**Example** **10.**
*Let us continue the analysis of Example 9. Function g:[0,1]→[0,1], $g\left(p\right)=\sin^{2}\frac{\pi }{2}p$, is one-to-one. It can be continued to the bijection g:R→R by the periodic repetition,*
(88)$$\begin{array}{c} g\left(x\right)=n+\sin^{2}\frac{\pi }{2}\left(x-n\right),\;n\leq x\leq n+1,\;n\in Z. \end{array}$$

*Now let $f=g^{-1}$. (Equation 88) leads to a non-Diophantine arithmetic and non-Newtonian calculus. Let θ=α−β, 0≤θ≤π, be an angle between two vectors representing directions of Stern-Gerlach devices. Quantum conditional probability (Equation 87) can be represented in a non-Newtonian hidden-variable form,*
(89)$$\begin{array}{ccc} \cos^{2}\frac{\alpha -\beta }{2} & = & \sin^{2}\frac{\pi }{2}\frac{\pi -(\alpha -\beta )}{\pi }=f^{-1}\left(\frac{1}{\pi }\int_{\alpha }^{\pi +\beta }dr\right)=f^{-1}\left(\int_{f(\alpha^{\prime })}^{f(\pi^{\prime }⊕\beta^{\prime })}\tilde{\rho }\left(r\right)dr\right)\\ & = & \int_{\alpha^{\prime }}^{\pi^{\prime }⊕\beta^{\prime }}\rho \left(\lambda \right)D\lambda , \end{array}$$
*where $x^{\prime }=f^{-1}\left(x\right)$. Here, ρ is a conditional probability density of non-Newtonian hidden-variables (the half-circle is a result of conditioning by the first measurement).*


Non-Newtonian calculus shifts the discussion on relations between classical and quantum probability, or classical and quantum information, into unexplored areas.

**Example** **11.**
*In typical Bell-type experiments one deals with four probabilities, corresponding to four combinations (±,±), (±,∓) of pairs of binary results. The corresponding non-Newtonian model is obtained by rescaling $g\left(p_{k}\right)\mapsto p\;g\left(p_{k}/p\right)$, with p=1/2. The rescaled bijection satisfies $g\left(p_{1}\right)+g\left(p_{2}\right)=p$ for any $p_{1}+p_{2}=p$. Explicitly,*
(90)$$\begin{array}{c} g\left(p_{++}\right)+g\left(p_{+-}\right)+g\left(p_{-+}\right)+g\left(p_{--}\right)=1=p_{++}+p_{+-}+p_{-+}+p_{--}. \end{array}$$

*The resulting hidden-variable model is local, but standard Bell’s inequality cannot be proved [15]. Why? Mainly because the non-Newtonian integral is not a linear map with respect to the ordinary Diophantine addition and multiplication (unless f is linear), whereas the latter is always assumed in proofs of Bell-type inequalities.*


A generalization to arbitrary probabilities, $p_{1}+\cdots +p_{n}=1$, leads to an affine deformation of arithmetic, an analogue of Benioff number scaling [21,22,23,24,25]. Affine transformations do not affect Kolmogorov–Nagumo averages.

**Lemma** **2.**
*Consider probabilities $p_{1},\ldots ,p_{n}$, n≥3. $g(p_{k})$ are probabilities for any choice of $p_{k}$ if and only if $g\left(p_{k}\right)=\frac{1-a+2ap_{k}}{n+(2-n)a}$, −1≤a≤1.*


**Proof.** See Appendix C. □

The bijection *g* implied by Lemma 2 depends on *n*. In infinitely dimensional systems, that is when *n* can be arbitrary, the only option is a=1 and thus g(p)=p is the only acceptable solution. However, in spin systems there exits an alternative interpretation of this property: The dimension *n* grows with spin in such a way that $g_{n}\left(p\right)\to p$ with n→∞ is a correspondence principle meaning that very large spins are practically classical. The transition non-Diophantine → Diophantine, non-Newtonian → Newtonian becomes an analogue of non-classical → classical.

**Example** **12.**
*Limitations imposed by Lemma 2 can be nevertheless circumvented in various ways. For example, let g(1)=1 for a solution g from Lemma 1, so that $1_{X}=1$. Obviously,*
(91)$$\begin{array}{c} 1=1_{X}⊙\cdots ⊙1_{X}=1⊙\cdots ⊙1=1\cdot \;\ldots \;\cdot 1. \end{array}$$

*Replacing each of the *1*s by an appropriate sum of binary conditional probabilities*
(92)$$\begin{array}{c} 1=g\left(p_{k_{1}\ldots k_{n}1}\right)+g\left(p_{k_{1}\ldots k_{n}2}\right)=g\left(p_{k_{1}\ldots k_{n}1}\right)⊕g\left(p_{k_{1}\ldots k_{n}2}\right) \end{array}$$
*we can generate various conditional classical or quantum probabilities typical of a generalized Bernoulli-type process, representing several classical or quantum filters placed one after another.*


## 8. Non-Newtonian Maximum Entropy Principle

Let us finally discuss the implications of our non-Newtonian form (Equation 32) of entropy for maximum entropy principles. Assume probabilities belong to X. Define the Massieu function [43] by
(93)$$\begin{array}{ccc} \Phi & = & S{⊖}_{Z}\alpha_{Z}{⊙}_{Z}N{⊖}_{Z}\beta_{Z}{⊙}_{Z}H, \end{array}$$
(94)$$\begin{array}{ccc} N & = & \underset{k}{⨁}{}_{Z}^{X}p_{k}=f_{Z}^{-1}\left(\sum_{k}f_{X}\left(p_{k}\right)\right), \end{array}$$
(95)$$\begin{array}{ccc} H & = & \underset{k}{⨁}{}_{Z}p_{k}{⊙}_{Z}^{XE}E_{k}=f_{Z}^{-1}\left(\sum_{k}f_{X}\left(p_{k}\right)f_{E}\left(E_{k}\right)\right), \end{array}$$
where $E_{k}\in E$, and $\alpha_{Z}=f_{Z}^{-1}\left(\alpha \right)$, $\beta_{Z}=f_{Z}^{-1}\left(\beta \right)$ are Lagrange multipliers. Explicitly,
(96)$$\begin{array}{c} \Phi =f_{Z}^{-1}\left[\sum_{k}f_{X}\left(p_{k}\right)\ln\left(1/f_{X}\left(p_{k}\right)\right)-\alpha \sum_{k}f_{X}\left(p_{k}\right)-\beta \sum_{k}f_{X}\left(p_{k}\right)f_{E}\left(E_{k}\right)\right]. \end{array}$$

Vanishing of the derivative of Φ,
(97)$$\begin{array}{c} \frac{D\Phi }{Dp_{l}}=0_{Z}, \end{array}$$
is equivalent to the standard formula for probabilities $f_{X}\left(p_{k}\right)$ (see the second form of non-Newtonian derivative in (Equation 18)),
(98)$$\begin{array}{c} \frac{d}{df_{X}\left(p_{l}\right)}\left(\sum_{k}f_{X}\left(p_{k}\right)\ln\left(1/f_{X}\left(p_{k}\right)\right)-\alpha \sum_{k}f_{X}\left(p_{k}\right)-\beta \sum_{k}f_{X}\left(p_{k}\right)f_{E}\left(E_{k}\right)\right)=0. \end{array}$$

Accordingly, the solution reads
(99)$$\begin{array}{ccc} p_{k} & = & f_{X}^{-1}\left(e^{-\beta f_{E}\left(E_{k}\right)}/\tilde{Z}\left(\beta \right)\right)=\mathrm{Exp}\left({⊖}_{E}\beta_{E}{⊙}_{E}E_{k}\right){⊘}_{X}Z_{X}\left(\beta \right), \end{array}$$
(100)$$\begin{array}{ccc} Z_{X}\left(\beta \right) & = & f_{X}^{-1}\left(\tilde{Z}\left(\beta \right)\right)=f_{X}^{-1}\left(\tilde{Z}\left(f_{E}\left(\beta_{E}\right)\right)\right)=Z\left(\beta_{E}\right), \end{array}$$
and involves the exponential function Exp:E→X we have encountered before. The normalization,
(101)$$\begin{array}{c} 1_{X}=\underset{k}{⨁}{}_{X}p_{k}=f_{X}^{-1}\left(\sum_{k}e^{-\beta f_{E}\left(E_{k}\right)}/\tilde{Z}\left(\beta \right)\right)=f_{X}^{-1}\left(1\right), \end{array}$$
implies the usual relation $\tilde{Z}\left(\beta \right)=\sum_{k}e^{-\beta f_{E}\left(E_{k}\right)}$.

Equivalently, directly at the level of X,
(102)$$\begin{array}{ccc} Z_{X}\left(\beta \right) & = & f_{X}^{-1}\left(\sum_{k}e^{-\beta f_{E}\left(E_{k}\right)}\right)=f_{X}^{-1}\left(\sum_{k}f_{X}\circ f_{X}^{-1}\left(e^{f_{E}\left({⊖}_{E}\beta_{E}{⊙}_{E}E_{k}\right)}\right)\right)\\ & = & f_{X}^{-1}\left(\sum_{k}f_{X}\left(\mathrm{Exp}({⊖}_{E}\beta_{E}{⊙}_{E}E_{k})\right)\right)=\underset{k}{⨁}{}_{X}\mathrm{Exp}\left({⊖}_{E}\beta_{E}{⊙}_{E}E_{k}\right)=Z\left(\beta_{E}\right). \end{array}$$

All the standard tricks one finds in thermodynamics textbooks will work here. For example,
(103)$$\begin{array}{ccc} H & = & f_{Z}^{-1}\left(\sum_{k}f_{X}\left(p_{k}\right)f_{E}\left(E_{k}\right)\right)=f_{Z}^{-1}\left(\sum_{k}e^{-\beta f_{E}\left(E_{k}\right)}f_{E}\left(E_{k}\right)/\tilde{Z}\left(\beta \right)\right)=f_{Z}^{-1}\left(-\frac{d\ln\tilde{Z}\left(\beta \right)}{d\beta }\right)\\ & = & {⊖}_{Z}f_{Z}^{-1}\left(\frac{d\ln\tilde{Z}\left(\beta \right)}{d\beta }\right)={⊖}_{Z}f_{Z}^{-1}\left(\frac{d\ln\tilde{Z}\left(\beta \right)}{df_{E}\left(\beta_{E}\right)}\right)={⊖}_{Z}f_{Z}^{-1}\left(\frac{d\tilde{A}\left(f_{E}\left(\beta_{E}\right)\right)}{df_{E}\left(\beta_{E}\right)}\right), \end{array}$$
for some function
(104)$$\begin{array}{c} \begin{array}{ccc} E & \overset{A}{⟶} & Z\\ f_{E}↓ & & ↓f_{Z}\\ R & \overset{\tilde{A}}{⟶} & R \end{array} \end{array}$$
we yet have to determine. Clearly,
(105)$$\begin{array}{ccc} \ln\tilde{Z}\left(\beta \right) & = & \tilde{A}\left(f_{E}\left(\beta_{E}\right)\right)=\tilde{A}\left(\beta \right),\\ A(x) & = & f_{Z}^{-1}\left(\tilde{A}\left(f_{E}\left(x\right)\right)\right)=f_{Z}^{-1}\left(\ln\left(\tilde{Z}\left(f_{E}\left(x\right)\right)\right)\right)=f_{Z}^{-1}\left(\ln\left(f_{X}\circ f_{X}^{-1}\left[\tilde{Z}\left(f_{E}\left(x\right)\right)\right]\right)\right) \end{array}$$
(106)$$\begin{array}{ccc} & = & f_{Z}^{-1}\left(\ln f_{X}\left(Z(x)\right)\right)=\mathrm{Ln}Z\left(x\right), \end{array}$$
where Z:E→X, Ln:X→Z. Ultimately,
(107)$$\begin{array}{ccc} H & = & {⊖}_{Z}\frac{D\left(\mathrm{Ln}\circ Z\right)\left(\beta_{E}\right)}{D\beta_{E}}. \end{array}$$

## 9. Final Remarks

Non-Newtonian calculus, and the non-Diophantine arithmetics behind it, are as simple as the undergraduate arithmetic and calculus we were taught at schools. Their conceptual potential is immense but they remain largely unexplored and unappreciated. Apparently, physicists in general do not feel any need of going beyond standard Diophantine arithmetic operations, in spite of the fact that the two greatest revolutions of the 20th century physics were, in their essence, arithmetic (i.e., relativistic addition of velocities and quantum mechanical addition of probabilities). It is thus intriguing that two of the most controversial issues of modern science—dark energy and Bell’s theorem—reveal new aspects when reformulated in generalized arithmetic terms.

One should not be surprised that those who study generalizations of Boltzmann–Gibbs statistics are naturally more inclined to accept non-aprioric rules of physical arithmetic. Anyway, the very concept of non-extensivity, the core of many studies on generalized entropies, is implicitly linked with generalized forms of addition, multiplication, and differentiation [54,59,60,61].

## Figures and Tables

**Figure 1 entropy-22-01180-f001:**
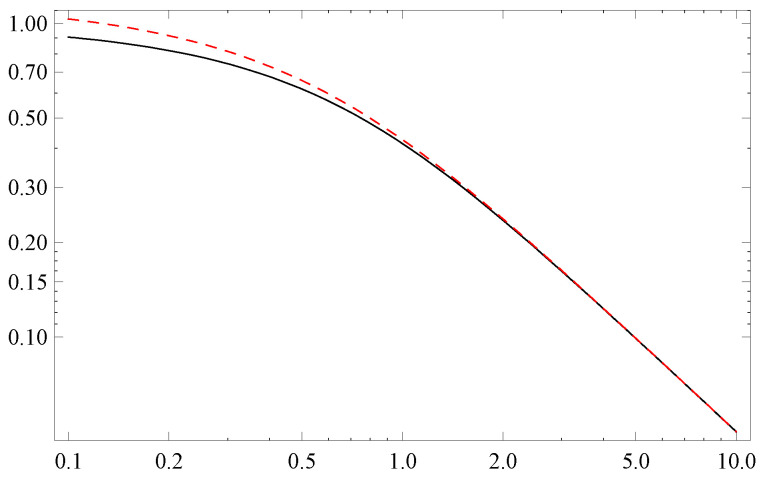
Log-log plots of Exp(−x) for $\kappa_{1}=1$, $\kappa_{2}=0$ (black), and $\kappa_{1}=\kappa_{2}=1$ (red). The tails are identical.

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
