# Peer review of "Unifying Aspects of Generalized Calculus"

_entropy, 2020, doi:10.3390/e22101180_

Round 1

Reviewer 1 Report

Please find attached report.

Author Response

Concerning the diagram for mixed arithmetics. I just added a comment that this type of arithmetic occurs in the chain rule for derivatives, and referred the reader to Example 6, where an example of a relevant diagram can be found.

Concerning notation. It's a question of taste and some efficiency, of course. I seriously contemplated the modifications suggested by the Referee, but finally I have decided to stick to the one I started with. Especially the last section would lose clarity, in my opinion, if I changed notation.

However, I have elaborated the thermodynamics aspects of the formalism in Example 6, and especially in the section on max-ent principles. Now I use the notion of a Massieu function, changed the sign before alpha, and worked out the issue of the statistical sum Z(beta). If I correctly understood the intention of the Referee, Eq. (107) is an example of a thermodynamic relation I've been asked to provide.

Reviewer 2 Report

This is an intriguing manuscript, which begins with quite a simple
observation of isomorphisms of ordinary (Diophantine) real number
arithmetics. Apparently, this observation is mathematically trivial
but, according to the author, it leads to interesting insights in
theoretical physics (thermodynamics and quantum physics in
particular). As a theoretical physics MSc, later working in
information theory and theoretical computer science, I can very
superficially appreciate the importance of this paper. However, the
particular quote from the paper struck my attention: "the two greatest
revolutions of the 20th century physics were, in their essence,
arithmetic (relativistic addition of velocities, quantum mechanical
addition of probabilities)". This is an interesting point of view,
which should be made central to this paper, given its content, rather
than presented in the conclusions.

As the author rightly observes, if two algebras are isomorphic then
we cannot tell which is the "original" and which is its "projection".
As I understand, however, we can tell apart different physical
theories, so they cannot be perfectly isomorphic in any rigorous
sense. I think that the paper would gain much more weight if the
author would elaborate more about the places where the isomorphisms of
physical theories do break as well. Where are the joints along which
the nature can be split? Being concrete, from my perspective, I would
like to learn more about the problems connected to the normalization
of probabilities and generalized nonextensive entropies by Renyi and
Tsallis. I also wonder how it could be linked with nonextensivity of
Shannon entropy, i.e., excess entropy researched by Crutchfield and
Feldman. I recommend a major revision of this paper to give the author
an opportunity to provide the missing comments.

Author Response

I have added a simple Example 1 showing that two mathematically isomorphic structures can describe different physics. Moreover, the examples of dark energy and Bell's theorem I mention also belong to this category. The problem has too many aspects to include them in the present paper, but many details can be found in my earlier publications which I quote (the issues such as: dimensionless vs. dimensional quantities, special relativistic effects, quantum mechanical effects, Bell inequalities, expanding Universe, psychophysical measurements...). Concerning the arithmetic aspect of special relativity, I have added Example 2. Some subtleties concerning normalization of probability appear in section on escort probabilities. I have concentrated on elaborating the thermodynamic aspects, especially the max-ent section. I also added some other examples, for instance the one showing that "minus x" is positive if one works with the arithmetic in R_+ defined by ln x.

Round 2

Reviewer 1 Report

My comments have been incorporated, thank you!

I therefore suggest accepting the manuscript for publication in Entropy.

Reviewer 2 Report

Comparing to my first reading of the manuscript, the author has added
a few more clarifying examples. It is now more clear how he
understands arithmetical isomorphisms, namely, as isomorphisms of just
a part of the respective complete mathematical theories of nature. For
instance, comparing special relativity and classical mechanics, the
isomorphism concerns adding of velocities but not necessarily other
parts of the theories, such as the second law. These aspects
definitely are worth elaborating in the future research. I understand
that this is beyond the scope of the present manuscript, which only
sketches a research program and provides illustrative examples.
From this point of view, this manuscript seems complete.

I am surprised, however, that I am the sole reviewer of this
manuscript and no working theoretical physicist was asked as a
referee. As a mathematician having only a master degree in
theoretical physics, I am not sure if I do not overrate the novelty of
the paper. But it definitely struck my attention. Since I am
interested in nonextensivity of entropy, I appreciate the developments
of Sections 6 and 7 and I would consider looking for non-Diophantine
arithmetics in information theory.